# Effect of Sirolimus vs. Everolimus on CMV-Infections after Kidney Transplantation—A Network Meta-Analysis

**DOI:** 10.3390/jcm11144216

**Published:** 2022-07-20

**Authors:** Sebastian Wolf, Verena S. Hoffmann, Florian Sommer, Matthias Schrempf, Mingming Li, Martin Ryll, Ulrich Wirth, Matthias Ilmer, Jens Werner, Joachim Andrassy

**Affiliations:** 1Department of General, Visceral and Transplant Surgery, University Hospital Augsburg, 86156 Augsburg, Germany; sebastian.wolf@uk-augsburg.de (S.W.); florian.sommer@uk-augsburg.de (F.S.); matthias.schrempf@uk-augsburg.de (M.S.); 2Institute for Medical Information Processing Biometry and Epidemiology (IBE), Ludwig-Maximilian’s University, 81377 Munich, Germany; vhoffmann@ibe.med.uni-muenchen.de; 3Department of General, Visceral and Transplant Surgery, Ludwig-Maximilian’s University, 81377 Munich, Germany; mingmingli590@gmail.com (M.L.); martin-ryll@gmx.de (M.R.); ulrich.wirth@med.uni-muenchen.de (U.W.); matthias.ilmer@med.uni-muenchen.de (M.I.); jens.werner@med.uni-muenchen.de (J.W.)

**Keywords:** CMV-infection, mTOR-inhibitor, calcineurininhibitor, renal transplantation, network metanalysis

## Abstract

(1) Background: Following renal transplantation, infection with cytomegalovirus (CMV) is a common and feared complication. mTOR-inhibitor (mTOR-I) treatment, either alone or in combination with calcineurininhibitors (CNIs), significantly reduces the CMV incidence after organ transplantation. As of now, there is no information on which mTOR-I, sirolimus (SIR) or everolimus (ERL), has a stronger anti-CMV effect. (2) Methods: The current literature was searched for prospective randomized controlled trials in renal transplantation. There were 1164 trials screened, of which 27 could be included (11,655 pts.). We performed a network meta-analysis to analyze the relative risk of different types of mTOR-I treatment on CMV infection 12 months after transplantation compared to CNI treatment. (3) Results: Four different types of mTOR-I treatment were analyzed in network meta-analyses—SIR mono, ERL mono, SIR with CNI, ERL with CNI. The mTOR-I treatment with the strongest anti-CMV effect compared to a regular CNI treatment was ERL in combination with a CNI (relative risk (RR) 0.27, confidence interval (CI) 0.22–0.32, *p* < 0.0001). The other mTOR-I therapy groups showed a slightly decreased anti-CMV efficacy (SIR monotherapy (mono): RR 0.35, CI 0.22–0.57, *p* < 0.001; SIR with CNI: RR 0.43, CI 0.29–0.64, *p* < 0.0001; ERL mono: RR 0.46, CI 0.22–0.93, *p* = 0.031). (4) Conclusions: The anti-CMV effect of both mTOR-Is (SRL and ERL) is highly effective, irrespective of the combination with other immunosuppressive drugs. Certain differences with respect to the potency against the CMV could be found between SRL and ERL. Data gained from this analysis seem to support that a combination of ERL and CNI has the most potent anti-CMV efficacy.

## 1. Introduction

Infection with cytomegalovirus (CMV) is the most common viral infection after transplantation and usually occurs during the first months after transplantation. The clinical presentation ranges from self-limiting, asymptomatic viremia to CMV syndrome with fever and leukopenia to severe invasive CMV disease (e.g., pneumonia, diffuse colitis, gastrointestinal ulcerations, pancreatitis, multiple organ failure) [1]. In addition to these direct effects, CMV infection can also lead to impaired graft function with increased rejection rates and an increased incidence of opportunistic infections [2]. The highest risk for manifest CMV infection (11–72%) is in the CMV-negative recipient with a CMV-positive donor (donor+/recipient- (D+/R−)). CMV reactivation can also occur in seropositive patients who require intensified immunosuppressive therapy. The prevalence for CMV seropositivity varies from 40–100%, and the incidence of CMV disease in D−/R−transplantation is less than 5% [3].

CMV prophylaxis can effectively prevent CMV infection. Antiviral therapy can be initiated only after CMV has been detected or—as is standard at most transplant centers—can be administered as universal prophylaxis for six months. With CMV prophylaxis, however, it should be noted that late CMV infections (after discontinuation of ganciclovir/valganciclovir) occur more frequently than with pre-emptive therapy [4]. In addition, CMV infections occur in up to 20% of patients despite prophylaxis, depending on the risk constellation. Finally, the side-effect profile of the drugs commonly used for this purpose must not be disregarded [5,6].

Interestingly, differences in CMV incidence exist with respect to the type of immunosuppressants. It is known that mTOR inhibitors (mTOR-I), sirolimus (SIR) and everolimus (ERL), significantly reduce CMV incidence compared to regular calcineurin inhibitor (CNI) therapy [7,8]. This beneficial effect exists when mTOR-I are used as the main immunosuppressant as well as in combination with a CNI [8]. However, it is not yet known if there are differences regarding anti-CMV potency between SRL and ERL.

## 2. Materials and Methods

This is a network meta-analysis of two different mTOR-Is (SIR or ERL, monotherapy or in combination with CNI) in comparison with CNI treatment from randomized clinical trials (RCTs) with respect to posttransplant CMV incidence. This meta-analysis is reported in accordance with the Preferred Reporting Items for Meta-Analyses (PRISMA-NMA) (Appendix A) [9].

### 2.1. Identification of the Eligibly Trials

The current literature was searched for prospective randomized controlled trials in renal transplantation. Retrospective data was excluded. Full reports were searched via PubMed (http://www.ncbi.nlm.nih.gov, accessed on 1 January 2021), ScienceDirect (http://www.sciencedirect.com, accessed on 1 January 2021) and the Cochrane Central Register of Controlled Trials (http://www.mrw.interscience.wiley.com/cochrane/cochrane_clcentral_articles_fs.html, accessed on 1 January 2021) up to December 2020 using the optimally sensitive strategies for the identification of eligible trials, combined with the following MeSH terms: (mTOR-inhibitor OR sirolimus OR everolimus) AND transplant AND (kidney OR renal).

### 2.2. Eligibility Criteria

Only prospective randomized renal transplantation trials published since 2004 (earliest recruitment starting 1998) were included. These trials were required to have at least two treatment arms, one with an mTOR-I-based immunosuppression (SIR or ERL) either with or without a CNI, and one arm containing an mTOR-I-free CNI-based immunosuppression. The mTOR-I had to be introduced within 3 months after transplantation. The retrieved trials were screened for information on posttransplant CMV infections 12 months after transplantation. For this purpose, cumulative CMV incidences were collected. When several publications showed the same cohort of patients, the information was summarized. The screening and inclusion of the articles was performed by two reviewers (S.W., J.A.).

### 2.3. Data Synthesis and Statistical Analysis

The included clinical trials were summarized qualitatively in tables describing the types of direct and indirect comparisons.

To summarize the available evidence, we calculated relative risks (RRs) for the incidence of CMV infections 12 months post transplantation under CNI- and mTOR-I-based immunosuppression. Accounting for possible heterogeneity, we performed a random effects network meta-analysis to compare the relative treatment effect of the different immunosuppressants. Standard errors were estimated using incidences and the number of patients per group. The meta-analysis was performed using statistical software package R (version 4.1.2) with the netmeta package (version 4.15-1) [10]. The network meta-analysis is used for the comparison of multiple treatments, as it performs direct comparisons between two trials (A vs. B) and indirect comparisons between trials with a common treatment (A vs. C, using trials comparing A vs. B and B vs. C) [11]. *p* values below 0.05 were considered significant, and all confidence limits were on the 95% level.

To visualize the evidence in the network, we used a network geometry graph. The network is laid out in the plane, where the nodes in the graph layout correspond to the treatments, and the edges display the observed treatment comparisons. The thickness of the edges is proportional to the number of participants in trials, directly comparing the four connected treatments.

The results of the network meta-analysis are reported as estimates of the RRs and visualized in a forest plot sorted to reflect the treatment ranking according to the analysis.

The within-design heterogeneity was assessed using Cochran’s Q test and Higgins’ I^2^ [12]. According to the values of I^2^, the heterogeneity will be considered as not important (0 to 40%), moderate (30 to 60%), substantial (50 to 90%), or considerable (75 to 100%) [13]. Possible network inconsistency was checked by comparing the estimated treatment effects via direct comparisons with those from the network analysis in a league table. The possible statistical significance of between-design inconsistency was assessed using the Q statistic and respective tests.

### 2.4. Data Extraction and Methodologic Quality

The following data were extracted from eligible articles by two reviewers (S.W., J.A.): type of transplanted organ, induction therapy, number of patients per treatment arm, type and dose of mTOR-I, start of mTOR-I treatment post transplantation, trough levels, follow-up period, description, incidence of events of CMV infections 12 months post transplantation and statistical analysis of the posttransplant CMV infections under mTOR-Is and CNIs both alone and in combination.

The methodological quality was assessed by three reviewers (S.W., J.A., V.S.H.) using the Cochrane Collaboration’s tool and ITT analysis [14,15].

## 3. Results

### 3.1. Included Trials

The literature search produced 1164 results, of which 27 trials met the inclusion criteria. Thus, a total number of *n* = 11,655 patients could be included (Figure 1). Of these trials, 13 RCTs used Sirolimus (SRL, Appendix A [16,17,18,19,20,21,22,23,24,25,26,27,28,29,30,31,32,33,34,35,36,37,38]) and 14 Everolimus (ERL, Appendix A [39,40,41,42,43,44,45,46,47,48,49,50,51,52,53,54,55,56]) as the mTOR-I. Mostly, the mTOR-I was introduced de novo or very early (within the first month; *n* = 22, 81.5%). The majority used either monoclonal or polyclonal antibodies as induction therapy (*n* = 23, 85.2%).

Of the RCTs using SIR, *n* = 6 compared a SIR monotherapy with CNI, and *n* = 7 compared SIR + CNI with CNI treatment. Two ERL trials compared monotherapy with CNI and *n* = 10 ERL + CNI with CNI treatment. There were two ERL trials containing three different treatment arms, ERL, ERL with CNI, and CNI alone. The geometry of the treatment network is shown in the network graph (Figure 2).

All of these trials delivered data on the incidence of CMV infections 12 months post transplantation.

### 3.2. Methodologic Quality

All 27 RCTs were considered to be of good methodological quality according to the Cochrane Collaboration’s tool. Almost all the RCTs used intention to treat (ITT) to analyze the data (26/27). Data quality measurements apply to primary as well as secondary parameters as CMV incidences.

### 3.3. Incidence of CMV Infections 12 Months Post Tx—Direct and Indirect Estimators

Compared with CNI, the network meta-analysis with random effects showed that all mTOR-Is could reduce the risk of CMV infection. The lowest relative risk was found for the combination therapy of ERL and CNI (RR 0.268, 95% CI [0.206, 0.349]; 10 RCTs). All other mTOR-I treatment options also showed significantly reduced relative risks: SIR monotherapy—RR 0.361, 95% CI [0.229, 0.569]; 6 RCTs; SIR + CNI—RR 0.434, 95% CI [0.299, 0.630]; 7 RCTs; ERL monotherapy—RR 0.488, 95% CI [0.278, 0.859]; 2 RCTs (Figure 3).

### 3.4. Exploration of Inconsistency and Heterogeneity

There was only a moderate yet insignificant within-design heterogeneity detected in the model (I^2^ = 29.9% [0.0%; 56.7%], Cochrane’s Q test *p*-value = 0.1194).

The net league table (Table 1) shows some deviations between direct estimates and network estimates for the CNI vs. ERL and ERL vs. ERL + CNI comparisons only. However, the between-design inconsistency was not significant (Q test *p*-value = 0.1439).

## 4. Discussion

This is a systematic review analyzing the impact of mTOR-Is vs. CNIs on CMV infections following renal transplantation. Analyses were performed as a network meta-analysis to estimate the anti-CMV effect of each mTOR-I. Data of 27 RCTs with *n* = 11,655 patients were included, making this analysis the largest of its kind on this topic. Infections occur most often in the early posttransplant period when multiple immunosuppressive drugs are administered at high concentrations. Therefore, only those RCTs that had the mTOR-I introduced de novo or up to three months were included.

It is known that the incidence of CMV infections is significantly reduced under mTOR-Is compared to CNIs, as shown in previously published reports [7,57,58,59]. The data of this network meta-analysis is in accordance with these results, showing a benefit from all different mTOR-Is’ regimens for the CMV incidence. The lowest relative risk for CMV infection could be found under the combination therapy of ERL and CNI (RR 0.27).

SIR is a macrolide antibiotic produced from *Streptomyces hygroscopicus*. ERL is a derivative of sirolimus and has an additional hydroxyethyl group at the C(40). Both agents bind to the same intracellular immunophilin as tacrolimus (FKBP12), but instead of inhibiting calcineurin, the drug-receptor complex then binds to proteins known as “mammalian targets of rapamycin” (mTOR). This inhibits a multifunctional serine-threonine kinase that prevents both DNA and protein synthesis, resulting in cell cycle arrest [60]. Nevertheless, ERL has a different tissue and subcellular distribution, different affinities to active drug transporters and drug metabolizing enzymes, as well as differences in drug-target protein interactions including a much higher potency in terms of interacting with the mTOR complex 2 (mTORC2) than SIR [61].

There are different possible mechanisms for the anti-CMV effect of mTOR-Is. mTOR-Is are known not only to suppress but also enhance certain immune reactions, such as memory T cell functions [59], the quantity and quality of virus-specific CD8+ memory cells T cells and memory precursor cells [59]. Another immune-stimulatory effect caused by the inhibition of mTOR is an increase of proinflammatory cytokines such as IL-12 and IL-1beta, while the anti-inflammatory cytokine IL-10 is suppressed [62]. Myelomonocytic cells are important for the persistence and spread of CMV [63]. In human macrophages, a sustained mTOR activation is mandatory for an efficient viral protein synthesis, especially during the late phase of the viral cycle [64]. The treatment of these cells with an mTOR-I abrogated CMV replication. In addition, an increased MHC antigen presentation via autophagy in monocytes/macrophages and dendritic cells and counteracting immunosuppressive effects of steroids have been reported by mTOR inhibition [62,65]. mTOR-Is can also lead to the inhibition of CMV mRNA translation by preventing the phosphorylation and inactivation of the translational repressor 4EBP1. It can thus bind the mRNA cap recognition protein eIF4E, preventing the formation of the eIF4F complex and thereby blocking translation [66].

Up to now, there has not been sufficient evidence of differences between SIR and ERL regarding their antiviral activity. One small prospective study on patients after heart transplantation showed fewer infections under ERL therapy [67]. This could not be confirmed by others [68,69]. For instance, a retrospective study of 409 kidney transplant patients showed no significant difference in the rate of drug discontinuation due to severe infection between ERL- or SIR-treated patients (ERL 2.3%, SIR 4.8%, *p* = 0.17) [70].

In our previous meta-analysis on the impact of mTOR-Is on overall infections after renal transplantation, we could already show that the lowest risk for CMV infections was present under a combination therapy of mTOR-I and CNI [8]. With an RR of 0.27, the combination therapy of ERL and CNI in comparison to CNI confirms these results.

The following scenarios may serve as a potential explanation for our findings: Using the combination therapy, mTOR-I and CNI trough levels are substantially reduced. Perhaps, the positive antiviral effect of the mTOR-Is, even under the reduced dose, simply outweighs the additional immunosuppression of the combination therapy [71,72].

Why the combination of ERL and CNI is favorable compared to SIR and CNI and which molecular mechanism is responsible for this remains unclear. The same applies to the monotherapy of SIR and ERL. At present, a reliable clinical comparison of SIR and ERL is hampered due to the lack of well-controlled, prospective studies that directly compare SIR and ERL.

Our study has some limitations. There were some deviations between direct and network estimates; however, the between-design inconsistency was not significant. This difference between the network and the direct estimate for the ERL vs. ERL + CNI comparison can be explained by the low number of trials contributing to the direct estimator (*n* = 2), especially in the case of Chadban et al., with a low number of participants and events (ERL *n* = 49, ERL + CNI *n* = 30, number of events *n* = 2 in each arm). Another limitation is that, naturally, the primary endpoint in the included RCTs was survival and BPAR and not infection. This is reflected in inhomogeneous reporting on CMV parameters such as the type of antiviral prophylaxis, the virological method of CMV monitoring, the frequency of CMV surveillance and parameters for detecting clinically significant infection. However, the cumulative incidence of CMV infection is regularly recorded in most trials. This may have caused a certain bias originating from heterogeneity. In the absence of studies that directly compare the two different mTOR-Is, we had to perform a network meta-analysis, which for the first time was able to show the abovementioned differences.

## 5. Conclusions

Both mTOR-Is, SRL and ERL, exert their anti-CMV effect in mono- as well as combination therapy with a CNI. The data gained from the present analysis seem to support a possible superiority of the combination of ERL and CNI.

## Figures and Tables

**Figure 1 jcm-11-04216-f001:**
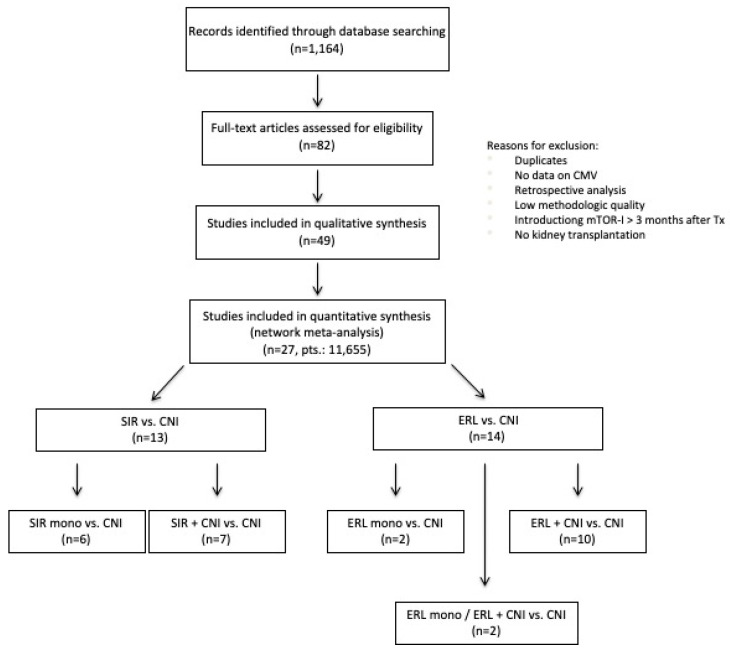
Flow chart of the selection of articles.

**Figure 2 jcm-11-04216-f002:**
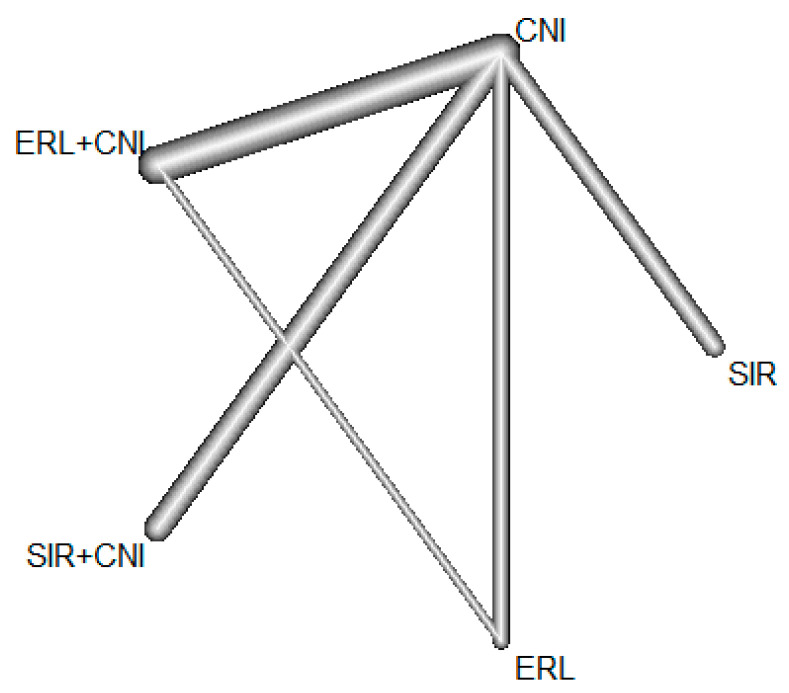
Geometry of the treatment network. The nodes in the graph layout correspond to the treatments, and the edges display the observed treatment comparisons. The thickness of the edges is proportional to the number of participants in trials, directly comparing the four connected treatments.

**Figure 3 jcm-11-04216-f003:**
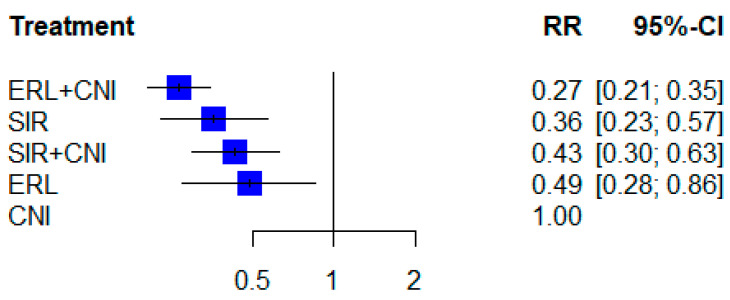
Forest plot indicating the relative risks (RR) of the occurrence of CMV infections on the four different mTOR-Is vs. CNI treatment post transplantation.

**Table 1 jcm-11-04216-t001:** League table (network estimates in lower triangle, direct estimates in upper triangle). Green: direct comparisons available, consistent with indirect network estimates. Orange: direct comparison available, but inconsistent with network estimate. Grey: network estimates available.

**CNI**	0.563 (0.314; 1.009)	0.434 (0.299; 0.630)	0.361 (0.229; 0.569)	0.265 (0.203; 0.346)
0.488 (0.278; 0.859)	**ERL**			1.067 (0.419; 2.716)
0.434 (0.299; 0.630)	0.888 (0.451; 1.748)	**SIR + CNI**		
0.361 (0.229; 0.569)	0.739 (0.358; 1.527)	0.832 (0.462; 1.499)	**SIR**	
0.268 (0.206; 0.349)	0.549 (0.301; 1.001)	0.618 (0.391; 0.976)	0.742 (0.439; 1.256)	**ERL + CNI**

## Data Availability

Not applicable.

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
