# Peer review of "Effect of Sirolimus vs. Everolimus on CMV-Infections after Kidney Transplantation—A Network Meta-Analysis"

_jcm, 2022, doi:10.3390/jcm11144216_

Round 1

Reviewer 1 Report

Please find file attached

Reviewer 2 Report

The manuscript by Wolf et al., represents a meta-analysis of Randomized Clinical Trials (RCT) comparing the effectiveness of mTOR inhibitors Sirolimus (SIR) or Everolimus (ERL) used alone or in combination with Calcineurininhibitors (CNIs) in reducing CMV incidence after renal transplantation.  The authors performed a literature search (PubMed, ScienceDirect, Cochrane Central Register of Controlled Trials) for RCT that met the Inclusion Criteria (described in Materials and Methods).  Using the extracted data, the authors calculated the relative risks (RRs) for the incidence of CMV infections post transplantation (12 months) for CNI- and mTOR-I-based (mono or combination) immunosuppression therapies.

In sum, this is a superb manuscript with no concerns about the methodologies used or the conclusions drawn from the data.  The results presented by this manuscript appear to be the largest of its kind, and accordingly, the conclusions (Lines 25-28: The anti-CMV 25 effect of both mTOR-Is (SRL and ERL) is highly effective irrespective of the combination with other immunosuppressive drugs, [and] a combination of ERL and CNI seems to be the most potent mTOR-I therapy against the CMV) are well-substantiated and represent an important advancement for the transplantation field vis-a-vis management of CMV infections.  There are no concerns about the methodologies, the inclusion/exclusion criteria, the statical analyses, or the conclusions.  

Author Response

Thank you very much for your kind review on our manuscript!

Round 2

Reviewer 1 Report

The majority of reviewer’s comments have been adequately delt with.  There is one small contradiction that would require rephrasing. At the end of the Absrtact as well as in point 5. Conclusions the statement about the superiority of the combination of ERL and CNI is too categoric. Taken all the limitations into account, it would be more appropriate to formulate that data gained from present analysis seem to support the possible superiority of this treatment modality (or any wording that would soften the originally too firm statement).
